# Association between epidural catheter tip malposition and anesthesiologists' experience after graduation: A cross-sectional study using postoperative CT images

**Mitsuhiro Matsuo**⬚\*, **Natsumi Sakamoto, Mariko Takebe, Tomonori Takazawa**

Department of Anesthesiology, University of Toyama, Toyama, Japan

\* m-matuo@toyama-nishi.jp

## Abstract

### Objectives

This study aimed to examine the incidence of epidural catheter tip malposition using postoperative CT images, and investigated its relationship with anesthesiologist and patient characteristics.

### Methods

Patients who had undergone epidural anesthesia at our hospital during the previous 18 years, and who had a thorax and abdominal CT scan within 5 days after surgery were included. Malposition was defined if the tip of the catheter did not penetrate the ligamentum flavum in postoperative CT images.

### Results

Among 189 eligible patients (median age 71 years, range 15–89), 78 (41%) were female. The median number of years of postgraduate experience of the physicians inserting the epidural catheter was 5.7 years (range 2.0–35.4). All epidural catheters were inserted using the paramedian approach in the left lateral decubitus position. The puncture site was the middle (48%) or lower (49%) thoracic spine. Epidural catheter malposition was observed in 24 patients (12.7%, 95% confidence interval [CI] 8.3–18.3). Among these cases, catheter tips were located at the vertebrae (vertebral arches: 9, transverse processes: 2, spinous process: 1), in superficial soft tissue (erector spinae: 5, subcutaneous: 4), and in deep soft tissue (intervertebral foramina: 2, subpleural space: 1). Anesthesiologists in the malposition group had significantly more experience since graduation (median 10.1 years vs. 5.6 years, P = 0.010). No other characteristics showed an association with catheter malposition.

**Data availability statement:** All relevant data are within the paper and its Supporting information files.

**Funding:** MM This work was supported by JSPS KAKENHI Grant Number JP21K08918. https://www.jsps.go.jp/j-grantsinaid/.

**Competing interests:** The authors have declared that no competing interests exist.

## Conclusions

Analysis of postoperative CT images revealed that the epidural catheter tip did not penetrate the ligamentum flavum in approximately 13% of cases. Our results suggest that even experienced anesthesiologists should be vigilant regarding proper catheter tip positioning.

## Introduction

Epidural anesthesia is a challenging procedure for anesthesiologists aiming to achieve adequate analgesia. Greater case experience is required for acquiring the skills for successful catheter placement in the epidural space compared to other essential skills needed by anesthesiologists, including orotracheal intubation and placement of arterial catheters [1]. Indeed, anesthesiologists with greater experience reportedly have higher initial successful puncture rates for neuraxial anesthesia [2,3]. Approximately 30% of epidural anesthesia cases result in inadequate postoperative analgesia, and are considered as failed procedures [4,5]. Anesthesiologists can implement strategies to improve the success of epidural anesthesia procedures by identifying factors that contribute to these failures. This study aimed to determine the incidence and factors contributing to epidural catheter tip malposition.

Postoperative CT images are useful tools for analyzing the trajectory of the epidural catheter from the skin to the ligamentum flavum [6]. Here, we used postoperative CT images to retrospectively examine the incidence of epidural catheter tip malposition, defined as cases in which the tip did not penetrate the ligamentum flavum. Further, we explored the factors contributing to epidural catheter tip malposition, including patient- and anesthesiologist-related characteristics.

## Methods

### Study design

This retrospective, cross-sectional study was conducted at Toyama University Hospital, a Japanese academic and tertiary care institution. The study was approved by the ethics committee of our hospital (Approval No. R2022221) on March 27, 2023, and complied with the principles of the Declaration of Helsinki. Since this was a retrospective study, the requirement for written informed consent was waived. Instead, an opt-out consent document was presented on our website for patients who did not wish to participate. Data was accessed for research purposes intermittently from April 2023 until June 2024. The data were analyzed anonymously. We followed the Strengthening the Reporting of Observational Studies in Epidemiology (STROBE) statement [7].

### Patient selection

We included all cases in which general anesthesia with epidural anesthesia was performed between January 1, 2005, and December 31, 2022. Among these patients,

those who underwent a chest and/or abdominal CT scan within five days after surgery, including the day of surgery, were identified. Patients with no visible catheter tips or duplicate cases were excluded. None of the patients declined study participation.

## Measurements

Data on patient characteristics, such as age, sex, height and weight, were retrieved from their electronic medical records. Information on unscheduled analgesic use within the first 24 hours after surgery was also extracted. Details of epidural anesthesia techniques, including patient positioning and insertion technique, were collected from their anesthesia records. The number of years of experience after graduation of the anesthesiologists was determined by calculating the number of days since graduation from medical school until the day the epidural anesthesia was administered. The vertebral level was analyzed from postoperative CT images, identifying the vertebral body where the catheter tip was located or the level just above the ligamentum flavum through which the catheter penetrated.

## Epidural anesthesia procedure

We used an 18-gauge Tuohy needle (B Braun, Tokyo) for performing epidural punctures, along with a radiopaque nylon epidural catheter (Smith Medical Japan, Tokyo) before 2012, and a radiopaque Perifix® catheter (B Braun, Tokyo) after 2012. Adhesive tape (Fix Kit-Epi®, ALCARE Co., Ltd., Tokyo) was used to secure the catheter to the skin. During surgery, continuous epidural infusion was initiated at the rate of 2–6 mL/h using a patient-controlled epidural analgesia pump (DIB PCA system II; DIB International Co., Ltd., Tokyo). Epidural local anesthetics included 0.2% ropivacaine or 0.25% levobupivacaine, with or without fentanyl at 2–5 µg/mL. Postoperatively, the infusion rate was adjusted as part of routine care under the surgeon's supervision.

## Epidural catheter tip position

Using postoperative CT images, we defined a normal catheter tip position in cases in which the epidural catheter tip penetrated the ligamentum flavum. Catheter tip malposition was defined when the tip did not penetrate the ligamentum flavum. If malposition was indicated, the location of the tip was determined from the CT images.

## Statistical analysis

Descriptive statistics are presented as frequencies (%) for categorical variables and medians [range] for continuous variables. Comparisons between groups were conducted using the chi-square and Mann-Whitney U tests. Logistic regression analysis was performed with catheter tip malposition as the dependent variable and number of postgraduate years as the independent variable. Results are reported with 95% confidence intervals (CI). A two-sided p-value of less than 0.05 was considered statistically significant. The receiver operating characteristic (ROC) curve defined the optimal cutoff value, as the value for which the Youden index (Youden index = sensitivity + specificity − 1) was maximized. Statistical analyses were conducted using EZR software, a graphical user interface for R (The R Foundation for Statistical Computing, Vienna, Austria) [8].

## Results

Between January 1, 2005, and December 31, 2022, 11,559 patients underwent combined general and epidural anesthesia at our hospital. Among them, 189 patients with postoperative CT images of the chest or abdomen with a visible epidural catheter tip were analyzed (Fig 1). All epidural catheters visible on CT images were radiopaque Perifix® catheters.

Table 1 summarizes the characteristics of the patients and the epidural anesthesia techniques. Median patient age was 71 years [15, 89] and 78 (41%) were female. Median body mass index (BMI) was 22.6 kg/m$^2$ [13.6, 35.2]. All epidural catheter insertions were performed using the paramedian approach in the left lateral decubitus position. Since almost all the

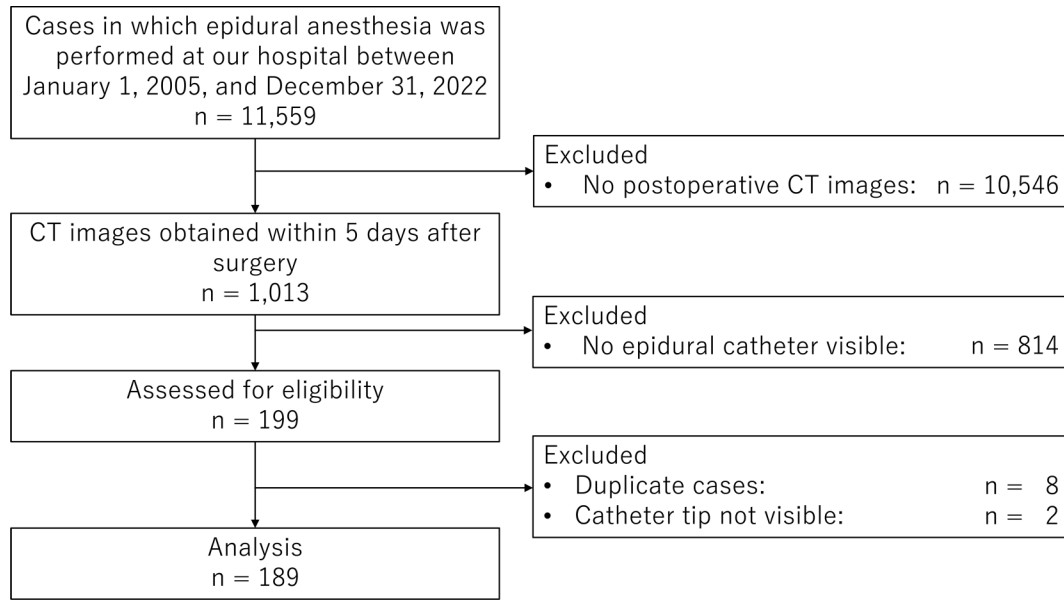

**Fig 1. Flow chart of study enrolment.**

**Table 1. Patient and procedural characteristics.**

| Variables | Value (n = 189) |
|---|---|
| Patient age, years | 71 [15, 89] |
| Patient sex, female | 78 (41%) |
| Patient body mass index, kg/m² | 22.6 [13.6, 35.2] |
| Insertion position | Left lateral decubitus: 189 (100%) |
| Insertion technique | Paramedian approach: 189 (100%) |
| Technique for identifying epidural space | Loss of resistance 189 (100%) |
| Vertebral level<br>T4/5/6/7/8/9/10/11/12/L1/2/3 | 1/2/6/34/49/41/23/18/10/4/0/1 |
| Anesthesiologists' experience, years | 5.7 [2.0, 35.4] |
| Anesthesiologist sex, female | 106 (56%) |
| Postoperative day 0/1/2/3/4/5 | 0/20/35/88/44/2 |
| Surgeon specialty, HBP/GI/Uro/ObGyn/Others | 110/39/31/6/3 |

The data are presented as frequencies (%) and medians [range]. Postoperative day indicates the day the CT image was taken. HBP, Hepatobiliary Pancreatic; GI, Gastrointestinal; Uro, Urology; ObGyn, Obstetrics and Gynecology.

CT scans were taken after abdominal surgery, most catheters (97%) were placed at the mid to lower thoracic level. In the 189 cases evaluated, epidural anesthesia was administered by 50 different anesthesiologists with a median of 5.7 years of postgraduate experience, ranging from 2.0 to 35.4 years (Fig 2).

Epidural catheter malposition, i.e., when the epidural catheter tip did not penetrate the ligamentum flavum, was found in 24 patients (12.7%, 95% CI 8.3–18.3%). In the malposition group, catheter tips were on the vertebrae in 12 cases (vertebral arches: 9, transverse processes: 2, spinous process: 1), in superficial soft tissue in nine cases (erector spinae: 5, subcutaneous: 4), and in deep soft tissue in three cases (intervertebral foramen: 2, subpleural space: 1) (Fig 3). Analyses

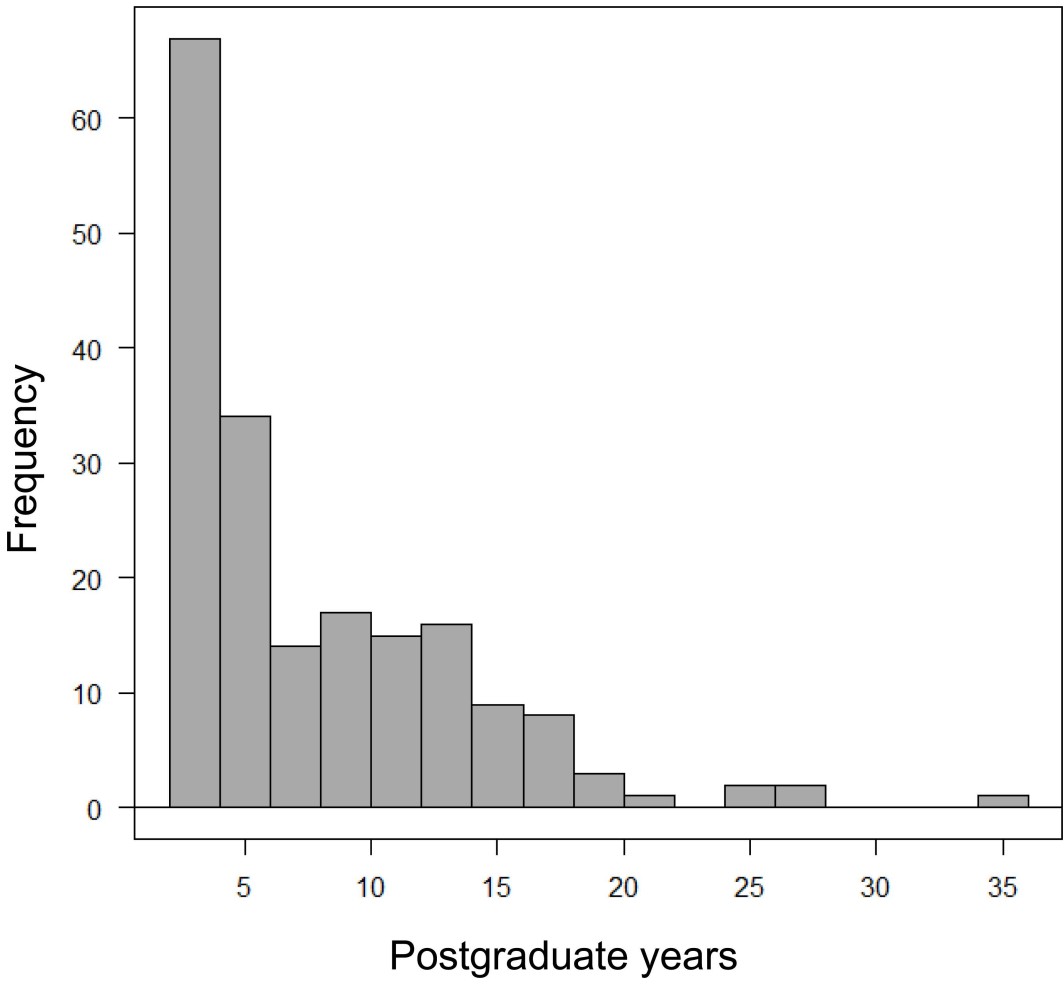

**Fig 2. Histogram of anesthesiologists' postgraduate experience.** The histogram shows the distribution of post-graduation years of experience of the anesthesiologists who performed the 189 epidural anesthesia procedures.

of the differences in characteristics between the normal catheter position and malposition groups indicated no significant differences in patient age, BMI, or puncture site. However, anesthesiologists in the malposition group had significantly more years of postgraduate experience (Table 2).

Unscheduled analgesic use within 24 hours postoperatively was significantly less frequent in the normal position group compared to the malposition group {68 (41%) vs. 17 (71%), P = 0.008}. The pain medications administered included pentazocine in 62 cases, non-steroidal anti-inflammatory agents in 11 cases, acetaminophen in 10 cases, and opioids in four cases.

Logistic regression analysis was conducted with catheter malposition as the dependent variable and postgraduate years as the independent variable. The results indicated a significantly greater incidence of malposition with greater anesthesiologist experience, with an odds ratio of 1.08 (95% CI 1.02–1.15) per postgraduate year.

ROC analysis performed to evaluate the relationship between the number of years post-graduation and catheter malposition (Fig 4) showed an area under the curve of 0.66 (95% CI 0.54–0.78). The optimal cutoff value was 11.3 years, with a specificity of 78%, sensitivity of 50%, positive likelihood ratio of 2.27, and negative likelihood ratio of 0.64.

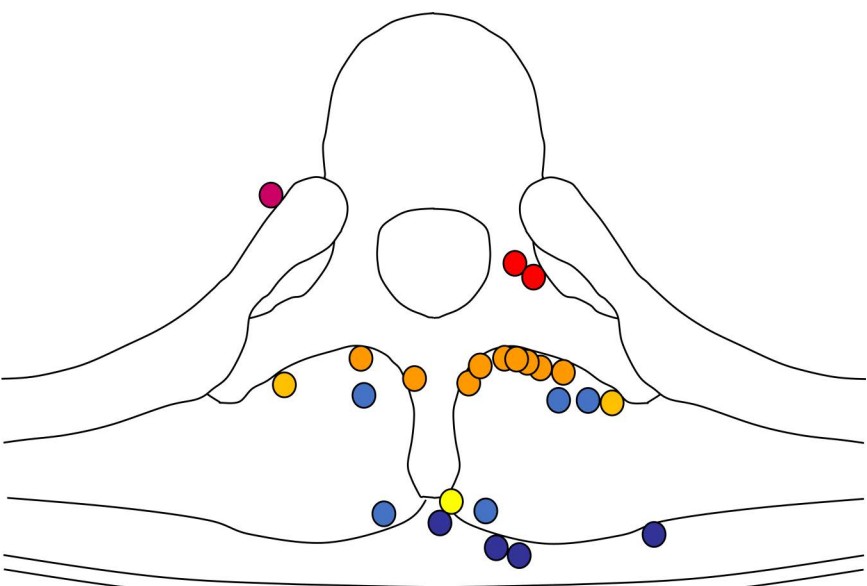

**Fig 3. Spatial distribution of epidural catheter tip malposition.** Epidural catheter malposition, defined when the epidural catheter tip did not penetrate the ligamentum flavum, was observed in 24 patients. The catheter tips were found at various locations: on the vertebral arches (orange), on the transverse processes (yellow), on the spinous process (bright yellow), in the spinal erector spinae (light blue), in the subcutaneous tissue (dark blue), and in the intervertebral foramen or the subpleural space (magenta).

**Table 2. Patient and procedural characteristics related to epidural catheter tip malposition.**

| | Normal position (n = 165) | Malposition (n = 24) | P value |
|---|---|---|---|
| Patient age, years | 71 [15, 89] | 70 [27, 86] | 0.497 |
| Patient sex, female | 69 (42%) | 9 (38%) | 0.825 |
| Patient body mass index, kg/m$^2$ | 22.9 [14.9, 35.2] | 21.3 [13.6, 31.3] | 0.286 |
| Vertebral level T4/5/6/7/8/9/10/11/12/L1/2/3 | 1/2/4/32/45/37/ 21/13/8/2/0/0 | 0/0/2/2/4/4/ 2/5/2/2/0/1 | 0.249 |
| Anesthesiologists' experience, years | 5.6 [2.0, 35.4] | 10.1 [2.1, 26.6] | 0.010 |
| Anesthesiologist sex, female | 93 (56%) | 13 (54%) | 0.830 |
| Postoperative day 0/1/2/3/4/5 | 0/17/30/75/41/2 | 0/3/5/13/3/0 | 0.253 |
| Length of epidural catheter advanced after LOR, cm | 5.0 [3.0, 7.0] | 5.0 [4.0, 6.0] | 0.622 |

The data are presented as frequencies (%) and medians [range]. Postoperative day indicates the day the CT image was taken. Comparisons between groups were conducted using the chi-square and Mann-Whitney U tests. LOR, loss-of-resistance.

We performed three sensitivity analyses to confirm the robustness of our results. For the first sensitivity analysis, one anesthesiologist who performed the most epidural anesthesia procedures (n = 13) and had the highest number of malpositions (n = 4) was excluded (S1 Table). Despite this exclusion, the number of postgraduate years of the anesthesiologists in the malposition group remained significantly higher (p = 0.006). Second, anesthesiologists were divided into groups based on half-year post-graduation intervals. The group with the most significant number of anesthesiologists was the 2.0–2.5 years' experience group (n = 37). After excluding this group, sensitivity analysis demonstrated that the malposition group still had significantly more years of postgraduate experience (p = 0.037) (S2 Table). Third, the significance remained even after excluding six epidural anesthesia procedures performed by anesthesiologists with more than 20 years of postgraduate experience (p = 0.024) (S3 Table).

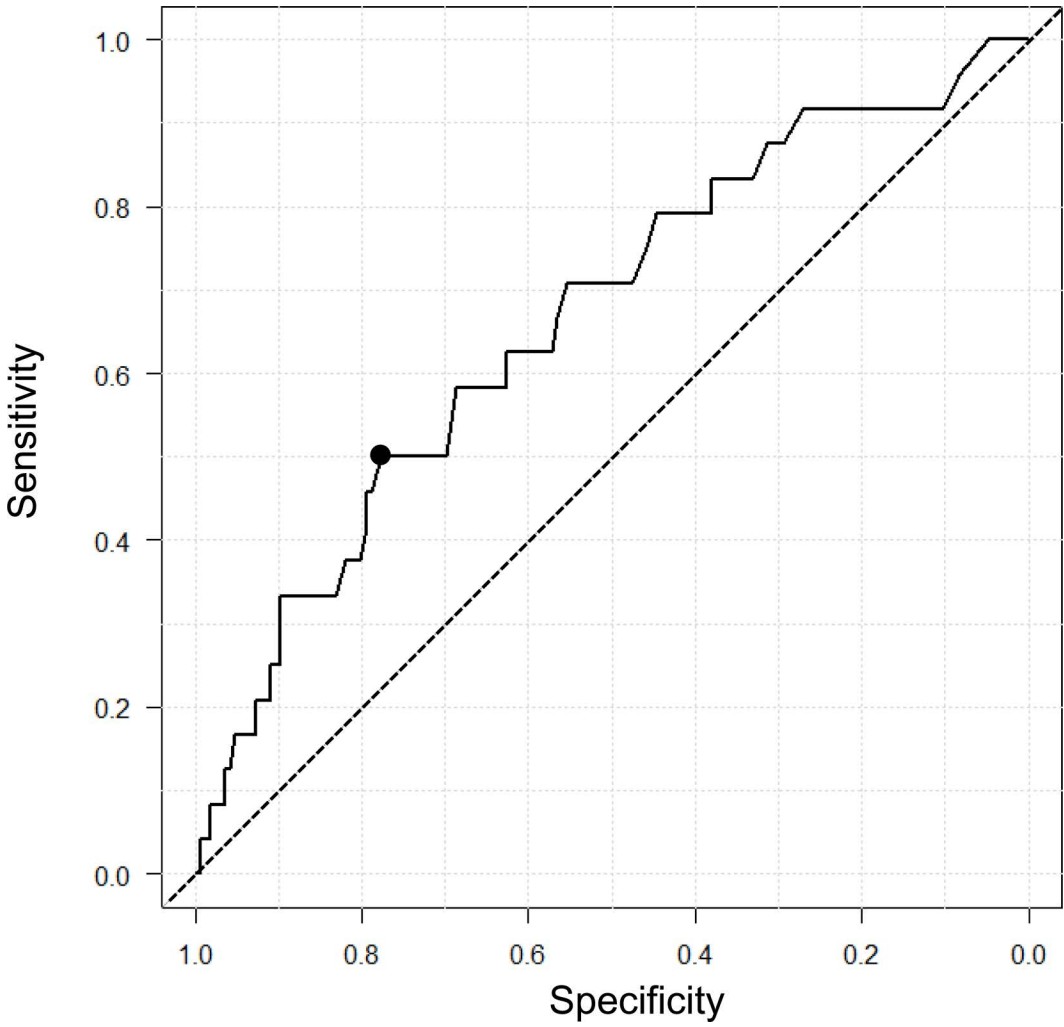

**Fig 4.** **Receiver operating characteristic curves of anesthesiologists' postgraduate experience for predicting epidural catheter tip malposition.**
The results showed an area under the curve of 0.66 (95% CI: 0.54-0.78). The optimal cutoff value, defined by the receiver operating characteristic curve as the value at which the Youden index (Youden index = sensitivity + specificity − 1) was maximized, was 11.3 years. At this cutoff, the specificity was 78%, and sensitivity was 50%, as shown by the closed circle. The dashed diagonal line represents the results of random guessing.

## Discussion

In this study, we used postoperative CT images to determine the incidence of epidural catheter tip malposition and the factors associated with malposition. Epidural catheter malposition was found in 24 patients (12.7%). Unexpectedly, the postgraduate experience of physicians was significantly greater in the malposition group. Logistic regression analysis showed that each additional post-graduation year was associated with a significantly higher incidence of malposition, with an odds ratio of 1.08 (95% CI 1.02–1.15).

The reason for the increased incidence of malpositioning with increasing years of experience is unclear. There are two possible causes of catheter tip malposition. The first is primary malposition due to failure of catheter placement. Attending anesthesiologists tend to have a lower rate of primary failure of epidural anesthesia than do trainees [9]. However, the increased failure rate could be related to carelessness and overconfidence associated with experience [10]. The second cause is secondary malposition due to dislocation resulting from patient movement after catheter placement. The epidural

catheter is reportedly withdrawn a few centimeters from the skin by patient movement [11]. Additionally, as anesthesiologists become more experienced, they might not pay as much attention to securing the catheter to the skin. Regardless of the causes of tip malposition, this study indicates that even experienced anesthesiologists need to pay attention to proper catheter tip positioning. Further, our results highlight the importance of continued vigilance and technique refinement, even among experienced anesthesiologists, to ensure optimal outcomes in epidural anesthesia.

Not only anesthesiologist-related factors, but also patient-related factors are important determinants of the success rate of epidural puncture. A previous prospective observational study showed that the rate of failed punctures is 3.0 times higher when the spinous process is not palpable [12]. In another prospective observational study, the success rate of initial puncture for neuraxial anesthesia was dependent on palpability of the spinous process (odds ratio 1.92) and the ease of adequate patient positioning (odds ratio 3.84) [2].

This study has several limitations. First, as described above, it was impossible to determine whether the postoperative catheter malposition was primary or secondary. Second, penetration of the ligamentum flavum by the epidural catheter tip does not always provide adequate analgesia. The spread of contrast medium within the epidural space is a good indicator of anesthetic efficacy [13], and leakage of contrast medium from the epidural space is associated with inadequate anesthetic outcomes [14]. Third, we used the number of postgraduate years as a marker of anesthesiologists' experience, but did not account for the number of epidural anesthesia cases performed since graduation. Fourth, because our institution is a teaching hospital, the median years of experience was relatively low, at 5.7 years. Fifth, due to a lack of data, we could not account for the patient's body habitus, including factors such as the ease of palpating the spinous process and positioning the patient.

## Conclusions

Epidural catheter tip malposition determined by CT taken postoperatively occurred in approximately 13% of the study cases. The greater the number of years since graduation of the anesthesiologist who inserted the epidural catheter, the more likely was the epidural catheter tip to be malpositioned. Even anesthesiologists with many years of experience need to pay careful attention to proper catheter tip positioning.

## Supporting information

**S1 Table. Sensitivity analysis excluding one anesthesiologist who performed the most epidural anesthesia procedures and had the highest number of malpositions.**
(DOCX)

**S2 Table. Sensitivity analysis excluding 37 epidural anesthesia procedures performed by anesthesiologists with 2.0–2.5 years' experience.**
(DOCX)

**S3 Table. Sensitivity analysis excluding six epidural anesthesia procedures performed by anesthesiologists with more than 20 years of postgraduate experience.**
(DOCX)

**S1 Dataset. The minimal data set used for analyzed.**
(XLSX)

## Acknowledgments

We thank Dr. Kazuma Nishikawa and Mr. Kazuaki Arai for assisting with CT data acquisition, and Mr. Toshio Fujimori for analyzing the epidural catheter data. The authors thank FORTE Science Communications (https://www.forte-science.co.jp/) for English language editing.

## Author contributions

**Conceptualization:** Mitsuhiro Matsuo.

**Data curation:** Mitsuhiro Matsuo, Natsumi Sakamoto.

**Formal analysis:** Mitsuhiro Matsuo, Natsumi Sakamoto, Mariko Takebe, Tomonori Takazawa.

**Funding acquisition:** Mitsuhiro Matsuo.

**Investigation:** Mitsuhiro Matsuo, Natsumi Sakamoto, Mariko Takebe.

**Methodology:** Tomonori Takazawa.

**Project administration:** Mitsuhiro Matsuo, Natsumi Sakamoto.

**Supervision:** Mariko Takebe, Tomonori Takazawa.

**Validation:** Mariko Takebe.

**Writing – original draft:** Mitsuhiro Matsuo.

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
