## [Decision Letter · Decision Letter 0]

PONE-D-24-56351Association between epidural catheter tip malposition and anesthesiologists’ experience after graduation: a cross-sectional study using postoperative CT imagesPLOS ONE

Dear Dr. Matsuo,

Thank you for submitting your manuscript to PLOS ONE. After careful consideration, we feel that it has merit but does not fully meet PLOS ONE’s publication criteria as it currently stands. Therefore, we invite you to submit a revised version of the manuscript that addresses the points raised during the review process.

We look forward to receiving your revised manuscript.

Kind regards,

Alessandro De Cassai, MD

Academic Editor

PLOS ONE

Journal Requirements:

 “MM

This work was supported by JSPS KAKENHI Grant Number JP21K08918.

https://www.jsps.go.jp/j-grantsinaid/”

3. We note that your Data Availability Statement is currently as follows: All relevant data are within the manuscript and in Supporting Information files.

Reviewers' comments:

Reviewer's Responses to Questions

**Comments to the Author**

1. Is the manuscript technically sound, and do the data support the conclusions?

Reviewer #1: Yes

Reviewer #2: Yes

2. Has the statistical analysis been performed appropriately and rigorously? 

Reviewer #1: Yes

Reviewer #2: I Don't Know

3. Have the authors made all data underlying the findings in their manuscript fully available?

Reviewer #1: No

Reviewer #2: Yes

4. Is the manuscript presented in an intelligible fashion and written in standard English?

Reviewer #1: Yes

Reviewer #2: Yes

5. Review Comments to the Author

Reviewer #1: Although the manuscript is an original research, it has a few deficiency. Two things mentioned in the results were not included in the figure;

1) sensitivity analysis after excluding anesthesiologists who performed the most epidural anesthesia and had the highest malposition rate

2) sensitivity analysis after excluding anesthesiologists with the highest range of experience

The information given in the manuscript should also be included in the results.

Reviewer #2: Dear Authors,

A well written simple study with an interesting result on anesthesia providers. There are only a few studies investigating the malposition of epidural catheter tips and this retrospective study has the highest number of cases so far.

One negligible spelling mistake: page 4, line 4 reference in brackets not in the right position.

6. PLOS authors have the option to publish the peer review history of their article (what does this mean? ). If published, this will include your full peer review and any attached files.

**Do you want your identity to be public for this peer review?** For information about this choice, including consent withdrawal, please see our Privacy Policy .

Reviewer #1: No

Reviewer #2: No

---

## [Author Response · Author response to Decision Letter 1]

30 Apr 2025

Response to Reviewers

Manuscript ID: PONE-D-24-56351

Title: Association between epidural catheter tip malposition and anesthesiologists’ experience after graduation: a cross-sectional study using postoperative CT images

Dear Dr. Alessandro De Cassai and Reviewers,

We would like to express our sincere appreciation for your valuable feedback and the opportunity to revise our manuscript. We carefully reviewed each comment and revised the manuscript accordingly. We believe that the quality and clarity of the manuscript have improved substantially through this process.

Below, we provide our point-by-point responses to all reviewer and editorial comments. All changes made to the manuscript are indicated in the file titled Revised Manuscript with Track Changes.

Reviewer #1

Comment:

Although the manuscript is an original research, it has a few deficiency. Two things mentioned in the results were not included in the figure;

1) sensitivity analysis after excluding anesthesiologists who performed the most epidural anesthesia and had the highest malposition rate

2) sensitivity analysis after excluding anesthesiologists with the highest range of experience

The information given in the manuscript should also be included in the results.

Response:

Thank you for pointing out this inconsistency. We have now included the results of both sensitivity analyses in new Supplementary Tables (S1,S3 Tables) rather than in a figure.

Regarding the second analysis, we found that the anesthesiologist with the longest experience (35.4 years) had performed only one epidural anesthesia case. Therefore, instead of excluding only this one case, we conducted a sensitivity analysis by excluding all six cases performed by anesthesiologists with ≥20 years of experience. These results are now presented in S3 Table, and the corresponding description in the Results section has been updated accordingly.

We have revised Figure 2 to display a histogram with postgraduate years grouped in 2-year intervals, which improves the clarity of the distribution of epidural procedures performed by anesthesiologists. Another sensitivity analysis, excluding procedures performed by anesthesiologists with 2.0-2.5 years’ experience, is shown in S2 Table.

Reviewer #2

Comment:

One negligible spelling mistake: page 4, line 4 reference in brackets not in the right position.

Response:

Thank you for pointing this out. We corrected the placement of the reference brackets on page 4, line 4. In addition, we reviewed all other in-text citations throughout the manuscript to ensure that reference numbering and bracket formatting are consistent and accurate.

Editorial Requirements

1. Formatting and File Naming:

We have revised the manuscript and file names to conform to PLOS ONE’s formatting and style requirements.

2. Funding Statement:

We have added the following sentence to clarify the role of the funders in the cover letter:

3. Data Availability:

We confirm that all raw data required to replicate the results of our study are now provided as Supporting Information (S1 Dataset). This includes:

• Data values underlying means and standard deviations

• Raw data used in statistical analyses

We believe this dataset fulfills the PLOS ONE requirement for a minimal data set.

We also re-examined the statistical outputs and found that there were numerical inaccuracies in four specific locations that are not essential to the main conclusions of the manuscript (Lines 163–164, 174, 188, and 191–193). These corrected values are fully traceable and included in the Supporting Information file titled “S1 Dataset.” We carefully verified that all other numerical values in the manuscript are accurate.

4. Reference List Review:

We have reviewed all references and confirm that none of the cited papers have been retracted.

We are grateful to the reviewers and the academic editor for their constructive and insightful comments. We hope that the revised manuscript now meets the standards required for publication in PLOS ONE.

Sincerely,

Mitsuhiro Matsuo, MD, PhD

---

## [Decision Letter · Decision Letter 1]

Association between epidural catheter tip malposition and anesthesiologists’ experience after graduation: a cross-sectional study using postoperative CT images

PONE-D-24-56351R1

Dear Dr. Matsuo,

We’re pleased to inform you that your manuscript has been judged scientifically suitable for publication and will be formally accepted for publication once it meets all outstanding technical requirements.

Kind regards,

Alessandro De Cassai, MD

Academic Editor

PLOS ONE

Additional Editor Comments (optional):

Manuscript is now acceptable in its current form

Reviewers' comments:

Reviewer's Responses to Questions

**Comments to the Author**

1. If the authors have adequately addressed your comments raised in a previous round of review and you feel that this manuscript is now acceptable for publication, you may indicate that here to bypass the “Comments to the Author” section, enter your conflict of interest statement in the “Confidential to Editor” section, and submit your "Accept" recommendation.

Reviewer #2: (No Response)

2. Is the manuscript technically sound, and do the data support the conclusions?

Reviewer #2: Yes

3. Has the statistical analysis been performed appropriately and rigorously? 

Reviewer #2: I Don't Know

4. Have the authors made all data underlying the findings in their manuscript fully available?

Reviewer #2: Yes

5. Is the manuscript presented in an intelligible fashion and written in standard English?

Reviewer #2: Yes

6. Review Comments to the Author

Reviewer #2: (No Response)

7. PLOS authors have the option to publish the peer review history of their article (what does this mean? ). If published, this will include your full peer review and any attached files.

**Do you want your identity to be public for this peer review?** For information about this choice, including consent withdrawal, please see our Privacy Policy .

Reviewer #2: No

---

## [Editor Report · Acceptance letter]

PONE-D-24-56351R1

PLOS ONE

Dear Dr. Matsuo,

I'm pleased to inform you that your manuscript has been deemed suitable for publication in PLOS ONE. Congratulations! Your manuscript is now being handed over to our production team.

Kind regards,

on behalf of

Dr. Alessandro De Cassai

Academic Editor

PLOS ONE